# A Quantitative Approach of Generating Challenging Testing Scenarios Based on Functional Safety Standard

Kang Meng [1,2,†], Rui Zhou [3,4,†], Zhiheng Li [1] and Kai Zhang [1,2,*]

1 Shenzhen International Graduate School, Tsinghua University, Shenzhen 518000, China
2 Research Institute of Tsinghua, Pearl River Delta, Guangzhou 510530, China
3 Institute of Systems Engineering, Macau University of Science and Technology, Macau 999078, China
4 Waytous Inc., Shenzhen 518000, China
* Correspondence: zhangkai@sz.tsinghua.edu.cn
† These authors contributed equally to this work.

**Abstract:** With the rapid development of intelligent vehicle safety verification, scenario-based testing methods have received increasing attention. As the space of driving scenarios is vast, the challenge in scenario-based testing is the generation and selection of high-value testing scenarios to reduce the development and validation time. This paper proposes a method for generating challenging test scenarios. Our method quantifies the challenges in these scenarios by estimating the risks based on ISO 26262. We formulate the problem as a Markov decision process and quantify the challenges in the current state using the three risk factors provided in ISO 26262: exposure, severity, and controllability. We then employ reinforcement learning algorithms to identify the challenges and use the state–action value matrix to select motions for a background vehicle to generate critical scenarios. The effectiveness of the approach is validated by testing the generated challenge scenarios using a simulation model. The results show that our method can ensure both accuracy and coverage, and the larger the state space is, the more accident-prone the generated scenarios are. Our proposed method is general and easily adaptable to other cases.

**Keywords:** testing scenario generation; functional safety; reinforcement learning

## 1. Introduction

With the advancements in artificial intelligence technology, many intelligent products, such as voice assistants on mobile phones, smart access control systems, convenient intelligent vehicles, and other devices, are appearing in our daily lives, improving the quality of human life. The need to validate their intelligence draws attention to these new artificial intelligence products. Li et al. [1] define artificial intelligence as follows: "Artificial intelligence is the intelligence exhibited by machines". This reveals the close link between artificial intelligence and intelligence testing. A practical assessment of intelligent vehicles' intelligence is required. An intelligent vehicle can only enter the market if it is accepted by society and legislators. On-road vehicle testing is well-known to be very expensive, time-consuming, and typically unrepeatable due to the testing scenarios and conditions. Hundreds of billions of miles may be needed to demonstrate the dependability of intelligent vehicles in terms of fatalities and injuries [2,3]. To effectively develop intelligent vehicles, many studies [4–6] develop and test them using modeling and simulation methods.

There are many discussions on the theory of safety assessment of intelligent vehicles [1,7,8]. Scenario-based testing is state of the art for intelligent vehicle safety verification. The greatest challenge in scenario-based testing is that road traffic is an open parameter space, in which an infinite number of possible traffic scenarios can occur [9]. As the traffic scenarios are dynamic and the space of these scenarios is vast, it is worth investigating how to generate and select high-value test scenarios to reduce the development and validation time.

First, we need to clarify the term "scenario". Some studies [10–12] introduce related terminology and define the scene as a temporal sequence of scenario elements, including the actions and events related to the participating elements that occur within this sequence. Some recent studies propose testing scenario generation methods based on different algorithms.

Combinatorial testing: Huang et al. [13] permutated and combined vehicles into possible test scenario groups, and generated test scenarios using scenario importance analysis based on the applicable scenarios of the main functions. Xie et al. [14] used a similar approach, comparing the relative position and movement relations between vehicles to generate a total possible test scenario group using the permutation and combination method. Hu et al. [15] proposed a method for autonomous vehicles based on combinatorial testing and Bayesian networks by selecting some parameters to describe the test scenarios, which were classified according to road types and driving tasks.

Importance sampling: Zhao et al. [16–18] proposed an accelerated evaluation to test automated vehicles and verified the method through car-following and lane-change scenarios. The importance sampling theory was used to ensure that the safety benefits of vehicles are accurately assessed under the accelerated tests, and the results show that the proposed techniques have great potential. Guo et al. [19] identified testing scenarios using the significance function established by the occurrence frequency of the scenario and the performance challenge between the driver and the vehicle. Wang et al. [20] built a hierarchical, structural model of the complex and diverse off-road scenarios and used importance sampling theory to determine test scenarios.

Control problem: Li et al. [21] formulated the process to search for critical scenarios as an optimal control problem, and proposed a method that can facilitate the design of test trajectories, which pursues the falsification of multiple requirements by a single trajectory through an appropriate formulation of this optimal control problem. Chou et al. [22] employed control synthesis to generate corner cases from controlled invariant sets and dual-game solutions.

Machine learning: Tian et al. [23] proposed a multi-objective search-based testing framework that constructs test scenarios using atomic maneuvers and motif patterns. They used a multi-objective genetic algorithm to search for adversarial and diverse test scenarios. Zhang et al. [24] used the analytic hierarchy process (AHP) method and the effect transmission model to construct a scenario space. Then, they used a support vector machine (SVM) to solve the scenario space's safety boundaries. Bayesian optimization was used to automate the process of generating adversarial self-driving scenarios that expose poorly engineered or poorly trained self-driving policies and increase the risk of collision with simulated pedestrians and vehicles [25]. Bayesian networks were used as the probabilistic models of the scenarios [15]. The chosen function was intended to determine the values of scenario parameters by taking into account both probability and frequency. The K-medoids algorithm was used to cluster and analyze trajectory data and obtained six typical crash scenarios between passenger cars and two-wheelers to construct high-risk test scenarios [26]. Xu et al. [27] proposed a scenario construction method for an automated driving functions field test. Seven clusters of basic road characteristics for the test scenarios were summarized and taken as the basic items to flexibly complete the scenario configuration. Tuncali et al. [28] presented an automated test generation approach that employs rapidly exploring random trees to explore boundary case scenarios in which an autonomous vehicle can no longer avoid a collision. Koschi et al. [29] proposed two novel falsification methods for detecting safety flaws in automated vehicle adaptive cruise control (ACC) systems. These methods employ rapidly exploring random trees to generate motions for a leading vehicle for the ACC under test to cause a rear-end collision.

Reinforcement learning: Lee et al. [30] proposed adaptive stress testing (AST), a scalable method that can search for the most likely state trajectory leading to an event. The approach uses a reinforcement learning formulation and solves it using Monte Carlo tree search (MCTS). Koren et al. [31] formulated the problem as a Markov decision process

and used reinforcement learning algorithms to find the most likely failure scenarios instead of using direct Monte Carlo sampling to find collision scenarios. Corso et al. [32] enhanced the AST by encoding domain-relevant information into the search procedure to discover a more significant and expressive subset of the failure space. Koren et al. [33] proposed a method for improving AST by employing a recurrent neural network that receives a set of initial conditions from a continuous space as input. Feng et al. [34,35] proposed a measure combining maneuver challenge and exposure frequency to search for critical scenarios. Qin et al. [36] used signal temporal logic (STL) to specify the target against which it is tested and the constraints that limit the reasonableness of the testing regime. They leveraged deep Q-learning algorithm to determine how to execute these behaviors.

As summarized above, this area is attracting much discussion and exploration at present. The above studies first preprocess natural or imprecise data, such as fuzzy logic [37–39], filtering key features, and precision processing. Then they use algorithmic models to analyze and learn scene features. Although the previous methods can generate and select challenging scenarios, they have some common shortcomings. These methods do not provide a quantitative assessment of scenarios and lack systematic industry guidance. As works [8,40] note, the central challenge for current intelligent testing is that most studies in the field have only focused on qualitative evaluation and lacked quantitative assessments. Furthermore, as these methods were not invented according to any popular standard in the automotive industry, it is not easy to generalize them to the markets or more cases, and also not easy to apply them in practice. Obtaining a quantitative assessment and reference standards for testing scenarios is indispensable to solving these problems. According to this knowledge, this paper presents a new method that quantifies the challenge in various scenarios by estimating the risk based on ISO 26262. The main contribution of our study is that we focus on quantification and use functional safety, a general standard in the automotive industry, as the quantification basis. We then use reinforcement learning (RL) [41] to formulate the problem to generate challenging scenarios and verify our method through simulation using two public datasets.

The rest of the paper is organized as follows. The elementary principle and design of the new approach is introduced in Section 2, and then we conduct experiments under car-following and cut-in cases in Section 3 and discuss the results in Section 4. Finally, we conclude the paper in Section 5.

## 2. The Elementary Principle and Design of the Method

There are no common standards for generating testing scenarios in the testing domain at present. Since testing involves safety, and there is an existing and highly operable safety standard called ISO 26262 [42] "Road vehicles—Functional Safety", a functional safety standard for automobiles developed by ISO that was compiled by car manufacturers, system suppliers, and automotive engineers [43], we introduced this standard to this problem on a theoretical basis. The specific quantification method was achieved through RL. The architecture of the proposed method is shown in Figure 1.

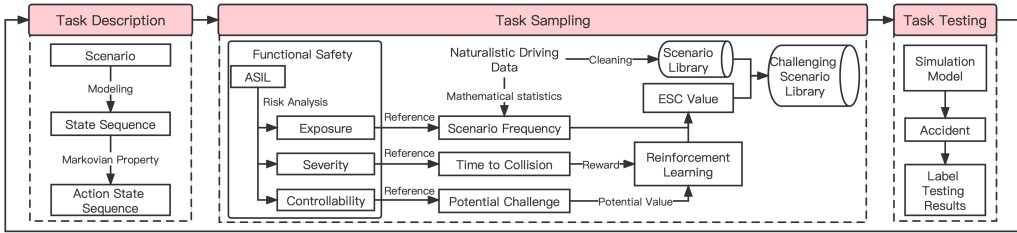

**Figure 1.** The system architecture of the proposed method.

### 2.1. Functional Safety: Theoretical Basis

ISO 26262 is an adaptation of the more general IEC 61508 standard for the automotive industry. According to ISO 26262, a hazard is a potential source of harm. Functional Safety

is defined as the "absence of unreasonable risk due to hazards caused by malfunctioning behavior of Electrical/Electronic systems". Part 3 of ISO 26262 is dedicated to hazard and risk analysis guidance, which determines the risk of harm/injury to people and property damage. Part 3 primarily introduces the requirements for risk analysis and risk assessment to determine the Automotive Safety Integrity Level (ASIL), a critical component of ISO 26262, and the level of risk reduction needed to achieve a tolerable risk is used to represent the stringency of the safety requirements [44].

ASILs, as previously stated, are essentially classification levels for hazardous events. In the event of a malfunction at the vehicle function level, a hazard and risk analysis is used to determine the ASIL. According to ISO 26262, an ASIL shall be determined for each hazardous event based on the classification of the probability of exposure, the severity, and the controllability, as shown in Table 1. Exposure is defined as the "state of being in an operational situation that can be hazardous if coincident with the failure mode under analysis". Severity is defined as the "estimate of the extent of harm to one or more individuals that can occur in a potentially hazardous event". Controllability is defined as "an ability to avoid a specified harm or damage through the timely reactions of the persons involved possibly with support from external measures". Each has corresponding levels. ASILs have five levels: QM, A, B, C, and D, where level D implies the highest safety requirement, level A means the least strict safety requirements, and level QM implies no special safety requirements [45].

**Table 1.** ASIL determination.

| Severity Class | Exposure Class | Controllability Class | | |
|---|---|---|---|---|
| | | C1 | C2 | C3 |
| S1 | E1 | QM | QM | QM |
| | E2 | QM | QM | QM |
| | E3 | QM | QM | A |
| | E4 | QM | A | B |
| S2 | E1 | QM | QM | QM |
| | E2 | QM | QM | A |
| | E3 | QM | A | B |
| | E4 | A | B | C |
| S3 | E1 | QM | QM | A |
| | E2 | QM | A | B |
| | E3 | A | B | C |
| | E4 | B | C | D |

As risk can be described as a function of the probability of exposure, the severity, and the controllability, based on ISO 26262, we introduced functional safety as our theoretical basis for testing scenarios. We aimed to quantify how risky scenarios are based on the three factors. The quantified value is defined as $\mathcal{ESC}$. Scenarios in which $\mathcal{ESC}$ exceeds the threshold are called challenging scenarios.

### 2.2. Quantification Method

In this section, we detail the specific method based on RL. The steps include the modeling scenarios part and $\mathcal{ESC}$ quantification algorithm part.

### 2.2.1. Scenario Modeling

RL is widely used in various areas, such as robotics, neuroscience, computer science, and automatic control. It introduces a way of programming agents by reward and punishment, without needing to specify how the task is to be achieved. Markov decision processes (MDPs) [41] are modeling basics of RL problems that usually include the following elements: S, A, $\mathcal{R}$, $\mathcal{T}$, and $\pi$.

- S is a set of states s of every agent.
- A is a discrete set of available agent actions that can be selected at a certain state.
- $\mathcal{R}$ is the reward function that represents the reward gained from taking a certain action at a specific state.
- $\mathcal{T}$ is the state transition probability matrix , which specifies the probability of taking action for a state. This is consistent with Markov Property, where the state transfer effect of taking action is only related to the current state, irrespective of the historical state.
- $\pi$ is the policy: S→A, where the goal of RL is to learn the optimal policy $\pi*$ to obtain the optimal expected reward.

In the following, we designate the vehicles we want to test as the host vehicles (HVs), and the other vehicles as the background vehicles (BVs). Based on [10–12], scenario $\mathbb{S}$ can be represented as a sequence of states, where subscripts indicate moments:

$$\mathbb{S} = (s_0, s_1, s_2, \cdots, s_k), \forall s \in S. \tag{1}$$

Due to the Markov Property:$(s_t, a_t) \to s_{t+1}$ [46], where $s_t$ is a current state and $s_{t+1}$ is the next state, a scenario $\mathbb{S}$ can also be represented as a sequence of initial state and actions in the following moments:

$$\mathbb{S} = (s_0, a_0, a_1, \cdots, a_{k-1}), \forall s \in S, \forall a \in A. \tag{2}$$

This is an adequate simplification. It not only takes into account vehicle-driving characteristics but also reduces the dimension of the scenarios, which helps to solve the bottleneck problem, known as the "dimension explosion", in testing scenario generation.

### 2.2.2. Challenging Scenarios

The level of risk in a testing scenario reflects the level of challenge in the scenario $\mathbb{S}$. We use $\mathcal{ESC}$ as a quantified value to represent the risk in a scenario according to ISO26262. As the states and actions that make up the scenarios have Markov Properties, $\mathcal{ESC}$ of a scenario $\mathbb{S}$ can also be represented as a sequence of units that reflect the $\mathcal{ESC}$ of states and actions:

$$\begin{aligned}
\mathcal{ESC}(\mathbb{S}) &= \mathcal{ESC}[(s_0, s_1, \cdots, s_k)] \\
&= \mathcal{ESC}(s_0) * \mathcal{ESC}(s_1) * \cdots * \mathcal{ESC}(s_k) \\
&= \mathcal{ESC}(s_0) * \mathcal{ESC}[(s_0, a_0)] * \cdots * \mathcal{ESC}[(s_{k-1}, a_{k-1})] \\
&= \mathcal{ESC}(s_0) * \prod_{i=0}^{k-1} \mathcal{ESC}[(s_i, a_i)].
\end{aligned} \tag{3}$$

The difficult scenarios were chosen based on *esc*. The *esc* of a scenario was compared to the threshold, with *esc* above the threshold being selected as challenging scenes and *esc* below the threshold being ignored:

$$\mathbb{S} = \begin{cases} challenging \quad scenarios & \text{if } \mathcal{ESC}(\mathbb{S}) > threshold \\ unchallenging \quad scenarios & \text{if } \mathcal{ESC}(\mathbb{S}) \leq threshold \end{cases} \tag{4}$$

As the $\mathcal{ESC}$ operation of a scenario can be seen as MDPs, the scenario generation process can be described as a decision tree, as shown in Figure 2, in which the red circles are valuable actions. A challenging scenario will select a red action as the next action.

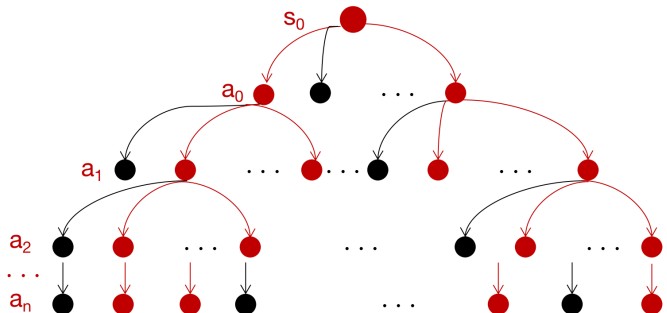

**Figure 2.** Scenario generating process can be described as a decision tree based on MDPs.

### 2.2.3. Quantification Analysis

Based on the previous analysis, the $\mathcal{ESC}$ of a state is determined by three factors: the probability of exposure $\mathcal{E}(s,a)$, severity $\mathcal{S}(s,a)$ and controllability $\mathcal{C}(s,a)$. Therefore, the three factors should be quantified to describe $\mathcal{ESC}[(s_i, a_i)]$ and $\mathcal{ESC}(s_0)$ in Equation (3).

- The value of $\mathcal{E}(s,a)$ is determined by naturalistic driving data (NDD) to reflect the probability of exposure. NDD describes the natural driving environment and can provide exposure information. By analyzing the samples in NDD, we can obtain the frequency of state as $\mathcal{Samples}(s)$ and the frequency of completing an action in the current state as $\mathcal{Samples}[(s,a)]$. Therefore, $\mathcal{E}(s,a)$ can be defined as:

$$\mathcal{E}(s,a) = P(a|s) = \frac{\mathcal{Samples}[(s,a)]}{\mathcal{Samples}(s)}. \tag{5}$$

- The value of $\mathcal{S}(s,a)$ should reflect the level of severity of completing an action a in a current state s. As the direct consequence of completing an action in the current state leads to the next state, $\mathcal{S}(s,a)$ can be quantified by analyzing the next state caused by the action. As the definition is similar to reward function $\mathcal{R}$, the $\mathcal{S}(s,a)$ can be seen as the reward $\mathcal{R}(s,a)$ in exploration. Time to collision (TTC) is regarded as a useful metric to obtain the driver perception of collision risk [47] and was selected to evaluate the forward collision warning (FCW) warning threshold time. TTC is defined as Equation (6) [48] where $D_r$ is the related distance and $V_r$ is the related velocity. We quantified the severity according to TTC, as shown in Equation (7).

$$TCC = \frac{D_r}{V_r}, \tag{6}$$

$$\mathcal{S}(s,a) = \mathcal{TTC}[(s,a)]. \tag{7}$$

- The definition of $\mathcal{C}(s,a)$ differs from that of ISO26262, because controllability aimed at hazardous events focuses on external measures after traffic accidents, whereas testing aimed at scenarios does not. However, the primary goal considers the potential consequences and long-term challenge effects of the current state. Based on this, we defined $mathcalC(s,a)$ as the expected $esc$ from actions in the next state to reflect the long-term value of actions in the current state, as shown in Equation (8).

$$\mathcal{C}(s,a) = E[\mathcal{ESC}[(s,a), \hat{a}]] = \frac{1}{m} \sum_{i=1}^{m} \mathcal{ESC}[(s,a), a_i]. \tag{8}$$

After the three factors are quantified, the $\mathcal{ESC}[(s_i, a_i)]$ and $\mathcal{ESC}(s_0)$ can be described by them:

$$\begin{aligned} \mathcal{ESC}[(s_t, a_t)] &= \mathcal{E}(s_t, a_t) \cdot \mathcal{S}(s_t, a_t) + \mathcal{C}(s_t, a_t) \\ &= P(a_t|s_t) \cdot \mathcal{TTC}(s_t, a_t) + \frac{1}{m} \sum_{i=1}^{m} \mathcal{ESC}[(s_{t+1}, a_i)], \end{aligned} \tag{9}$$

$$\mathcal{ESC}(s_0) = \mathcal{ESC}[(s_0, a_1)] + \mathcal{ESC}[(s_0, a_2)] + \cdots + \mathcal{ESC}[(s_0, a_m)] = \sum_{i=1}^{m} \mathcal{ESC}[(s_0, a_i)]. \quad (10)$$

### 2.2.4. $\mathcal{ESC}$ Algorithm

Based on RL, we propose using the *esc* algorithm to quantify scenarios through modeling and analysis. The algorithm's goal is the same as that of some popular RL algorithms, such as Q-learning and Sarsa, which work directly on the state-action value matrix $Q(S, A)$ to solve variants of the reinforcement learning problem [49,50]. The value of every unit $Q(s, a)$ of the matrix represents the discounted cumulative reward starting at state s, taking action a. The $\mathcal{ESC}$ algorithm, like Sarsa, is called on-policy learning, which selects actions at subsequent moments as the being same as the current moment. The updated law of the $\mathcal{ESC}$ values can be expressed as follows:

$$\mathcal{ESC}(s_t, a_t) \leftarrow \mathcal{ESC}(s_t, a_t) + \beta[(\mathcal{E}(s_t, a_t)\mathcal{S}(s_t, a_t) + \gamma\mathcal{C}(s_t, a_t)) - \mathcal{ESC}(s_t, a_t)], \quad (11)$$

where the meaning of $\mathcal{ESC}$ value is the same as the meaning of Q value. $\beta$ is learning rate, and $\gamma$ is discount factor. The equation can be further interpreted as:

$$\mathcal{ESC}(s_t, a_t) \leftarrow \mathcal{ESC}(s_t, a_t) + \beta\left[\left(P(a_t|s_t)\mathcal{TTC}(s_t, a_t) + \gamma\frac{1}{m}\sum_{i=1}^{m}\mathcal{ESC}[(s_t, a_t), a_i]\right) - \mathcal{ESC}(s_t, a_t)\right]. \quad (12)$$

The $\mathcal{ESC}$ algorithm is summarized in Algorithm 1. We reduce the complexity by matrix operations. Based on [51], Algorithm 1 requires $\mathbf{O}(|A||S|)$ memory and $\mathbf{O}(|A|^2|S|)$ computation per iteration. Furthermore, according to the $\mathcal{ESC}$ algorithm, the flow chart of our method is shown in Figure 3.

---

**Algorithm 1** $\mathcal{ESC}$ *Algorithm.*

---

Initialize $P(A|S), \mathcal{TTC}(S, A), \mathcal{ESC}(S, A)$;
    $\forall s \in S, a \in A, \mathcal{ESC}(s, a) = P(a|s)$;
    $error = Threshold + 1$;
**while** *error > Threshold* **do**
    Initialize $\mathcal{E\hat{S}C}(s, a) \leftarrow \frac{1}{m}\sum_{i=1}^{m}\mathcal{ESC}[(s, a), a_m]$;
    $D \leftarrow P(A|S) \cdot [\mathcal{TTC}(S, A) + \gamma\mathcal{E\hat{S}C}(S, A)]\text{-}\mathcal{ESC}(S, A)$;
    $\mathcal{ESC}(S, A) \leftarrow \mathcal{ESC}(S, A) + \beta D$;
    $error \leftarrow |D|$ ;
**end while**
**return** $\mathcal{ESC}(S, A)$

---

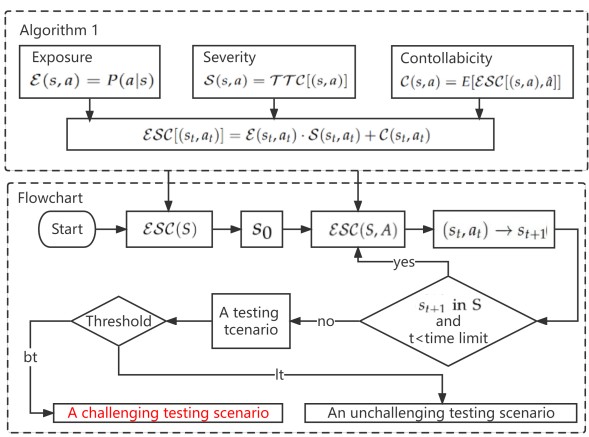

**Figure 3.** The flowchart of the proposed method.

## 3. Experiments

In this section, we first introduce the dataset and simulation verification model used in our experiments. We then conduct an experiment with two cases, the car-following case and the cut-in case, using two open-source datasets, HighD [52] and NGSIM [53], to test our algorithm.

### 3.1. Dataset

HighD [52], a large-scale, naturalistic, vehicle trajectory dataset, was collected from German highways. Based on a novel method to measure data from an aerial perspective, the dataset consists of 16.5 h of measurements from six locations with 110,000 vehicles, a total driven distance of 45,000 km, and 5600 recorded complete lane changes. As the positioning error is typically less than 10 cm and the frame rate used to record the video is 25 Hz, the dataset can provide precise and extensive experiment data.

NGSIM [53], the next-generation simulation, provides detailed, high-quality traffic datasets that aid in the modeling and simulation of intelligent algorithms to predict drivers' behavior and evade vehicular crashes [54]. Part of the US Highway 101 Dataset, consisting of 45 min of video, was collected from southbound US 101, also known as the Hollywood Freeway, and the study area was approximately 640 m. Eight synchronized digital video cameras, mounted from the top of a 36-story building adjacent to the freeway, recorded vehicles passing through the study area, providing data at a rate of 10 Hz.

### 3.2. Simulation Model

We introduced a simulation model, the intelligent driver model (IDM) [55] to validate the proposed novel method. IDM is a car-following model that works in mixed traffic. Based on [56], the model describes acceleration as a function of the gap $\Delta v_\alpha - L$, the speed $v_\alpha$, the desired velocity $v_0$ and the speed difference $\Delta v_\alpha$ between vehicle "$\alpha$" and the vehicle in front using the following expressions:

$$\frac{\mathrm{d}}{\mathrm{d}t} v_\alpha(t) = a \cdot \left( 1 - \left( \frac{v_\alpha}{v_0} \right)^\delta - \left( \frac{S^*(v_\alpha, \Delta v_\alpha)}{\Delta v_\alpha - L} \right)^2 \right), \tag{13}$$

where "$\delta$" is the acceleration component and the desired gap "$S^*$", which can be given as:

$$S^*(v_\alpha, \Delta v_\alpha) = S_0 + S_1 \cdot \sqrt{\frac{v_\alpha}{v_0}} + T \cdot v_\alpha + \frac{v_\alpha \cdot \Delta v_\alpha}{2\sqrt{a \cdot b}}. \tag{14}$$

where $S_0$ is the gap at jam conditions, $S_1$ is the gap factor, $T$ is the reaction of the driver, $a$ is maximum acceleration, and $b$ is minimum acceleration.

### 3.3. Car-Following Case

Figure 4 provides a basic illustration of a typical car-following case. Our previous introduction shows that we first modeled scenarios for specific applications. The states and actions of a testing scenario can be described based on the interaction between HV and BV. In the car-following case, we mainly focused on horizontal interactions. As the decision of BV is influenced by velocity $v$, relative velocity $v_r$, and relative distance $d_r$, a state is modeled as:

$$s = state = (v, v_r, d_r). \tag{15}$$

The action of the testing scenarios is the acceleration $acc$ of the BV. Therefore, the testing scenario is modeled as:

$$\mathbb{S} = (s_0, acc_0, acc_1, \cdots, acc_{k-1}), \tag{16}$$

where $s_0$ is the initial state and $acc_0, acc_1, \cdots, acc_{k-1}$ is the acceleration sequence of the BV. We obtained the distribution and density of these parameters after analyzing the HighD

and NGSIM, as shown in Figure 5. Then, in Table 2, the value range is determined. The discrete precision of velocity, distance, and acceleration was 1 m/s, 1 m, and 0.1 m/s$^2$:

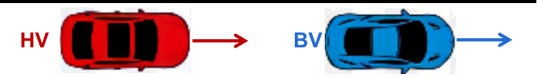

**Figure 4.** An illustration of car-following case. HV denotes a host vehicle. BV denotes a background vehicle.

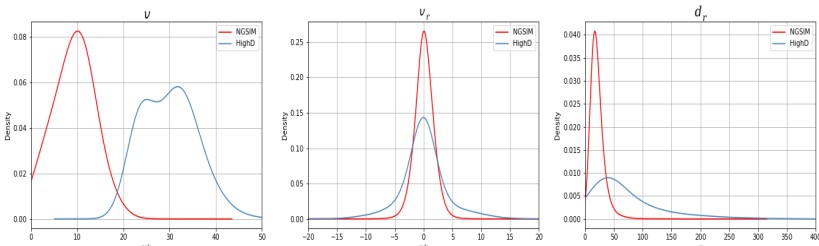

**Figure 5.** An illustration to show the distribution and density of parameters in the car-following case based on the HighD and NGSIM.

**Table 2.** Value range of parameters in car-following case.

| Parameter (HighD) | v (m/s) | $v_r$ (m/s) | $d_r$ (m) | a (m/s$^2$) |
|---|---|---|---|---|
| maximum | 50 | 17 | 391 | 2.2 |
| minimum | 20 | −18 | 4 | −3.0 |
| **Parameter (NGSIM)** | **v (m/s)** | **$v_r$ (m/s)** | **$d_r$ (m)** | **a (m/s$^2$)** |
| maximum | 29 | 20 | 210 | 3.4 |
| minimum | 0 | −19 | 2 | −3.4 |

The most extended scenario duration was designed to be 25 s, and the control frequency of the action was 1 Hz, which means that the BV selects an acceleration every second. As mentioned in the scenario modeling section, the MDPs-based modeling approach shrinks the parameter space. It is time-consuming to traverse the whole scenario to compute their $\mathcal{ESC}$ and find challenging scenarios with violent search algorithms. This study used the Markov property, $(s_t, a_t) \rightarrow s_{t+1}$, to choose the next action according to the current state. According to Table 2, the size of the space of states and action is:

$$size\_H(state) = 31 \times 36 \times 388 = 4.33 \times 10^5,$$
$$size\_H(action) = 53. \tag{17}$$

$$size\_N(state) = 30 \times 40 \times 209 = 2.51 \times 10^5,$$
$$size\_N(action) = 69, \tag{18}$$

where $size\_H$ means the data are from HighD, and $size\_N$ means the data are from NGSIM. Therefore, the size of the space of $\mathcal{ESC}$ matrix is $size(state)$ times $size(action)$:

$$size\_H(\mathcal{ESC}\ matirx) = 2.29 \times 10^7, \tag{19}$$

$$size\_N(\mathcal{ESC}\ matirx) = 1.73 \times 10^7. \tag{20}$$

Through $\mathcal{ESC}$ matrix and the threshold, pruning selection greatly reduces the search space for challenging scenarios. Based on our preprocessed data and modeling method, $\mathcal{E}(s,a)$ of the car-following case was designed as follows:

$$\mathcal{E}(s,a) = P(acc|s). \tag{21}$$

The result of $\mathcal{TTC}[(s,a)]$ can be used to calculate $mathcalS(s,a)$. The *esc* matrix was obtained after many iterations. Figure 6 depicts the *esc* of a state (30,3,10) using HighD. The green curve represents the esc value and the red curve represents $P(acc|s)$. When the threshold is set to zero, the figure shows that, when selecting the next BV action to generate challenging scenarios, more attention is paid to the BV's deceleration. This implies that both the frequency of occurrence and the level of risk are taken into account.

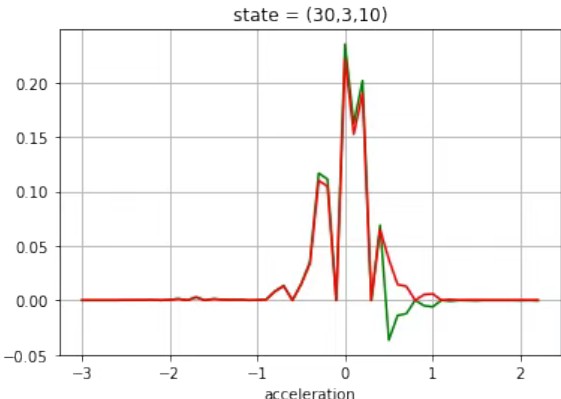

**Figure 6.** The $\mathcal{ESC}$ of state = (30,3,10) using HighD.

According to $\mathcal{ESC}$ matrix, we selected the initial state and action based on the threshold 0 and generated the testing scenario library. Then, we verified these generated scenarios by simulation experiments using the IDM model as the HV. If HV has a collision accident in a testing scenario, this scenario is genuinely challenging. The model weights are provided in Table 3, and the action space of BV is within the range provided in Table 2. The simulation result is shown in Figure 7.

**Table 3.** The weights of parameters in IDM in car-following case.

| Parameter | Weight_HighD | Weight_NGSIM | Interpretation |
|:---:|:---:|:---:|:---:|
| a | 2.2 | 3.4 | maximum acceleration |
| b | $-3$ | $-3.4$ | minimum acceleration |
| $v_0$ | 33 | 33 | desired velocity |
| $\delta$ | 4 | 4 | acceleration velocity |
| L | 4 | 2 | vehicle length |
| $S_0$ | 2 | 2 | gap at jam conditions |
| $S_1$ | 0 | 0 | gap factor |
| T | 1 | 1 | the reaction time of driver |

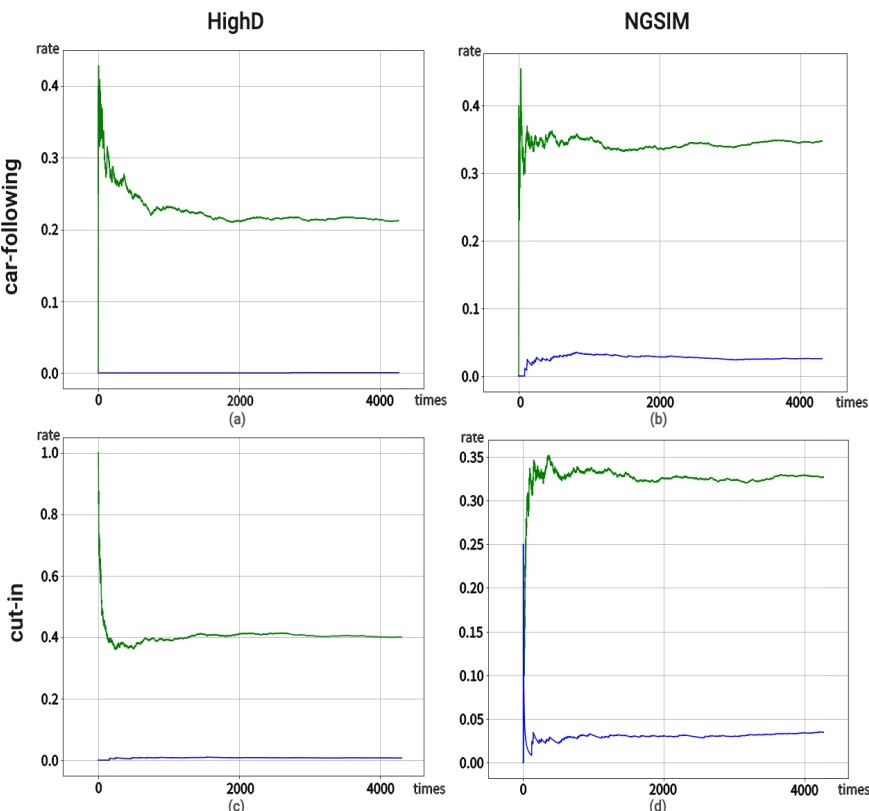

**Figure 7.** The simulation results of the car-following case and cut-in case using (**a**,**c**) HighD and (**b**,**d**) NGSIM datasets.

### 3.4. Cut-In Case

Figure 8 shows the primary illustration of a typical cut-in case. As in the previous experiment, we first modeled scenarios in specific applications based on the interaction between HV and BV. In the cut-in case, we focused on horizontal interactions and vertical interactions. There are six state variables in total which are velocity, relative velocity, and relative distance in both directions. However, since the longitudinal distance is short and the two datasets have different levels of detail in the longitudinal direction. HighD provides detailed motion statuses, such as distance, speed, and acceleration in the longitudinal direction, while NGSIM only provides a single distance in the longitudinal direction. So, the modeling and implementation of cut-in case experiments need to be adjusted and improved. We set the horizontal axis as the x-axis and the vertical as the y-axis. A state was modeled as follows:

$$s = state = (v^x, v_r^x, d_r^x, d_r^y) \tag{22}$$

The modeled state consists of relative velocity in x-axis $v^x$, relative velocity in x-axis $v_r^x$, relative distance in x-axis $d_r^x$, and relative distance in y-axis $d_r^y$. In many studies, longitudinal motion is simplified. For example, Feng et al. [57] simplified the longitudinal movement of the background vehicle by assuming that the longitudinal acceleration is 0. Our experiment used a relatively minor simplification. Because the longitudinal distance is short and only the longitudinal creative distance is in the state, the longitudinal velocity was used as the longitudinal action rather than the longitudinal acceleration. We need to further process the difference in longitudinal distance between two frames extracted from the NGSIM dataset to calculate the longitudinal speed. The action of testing scenarios was both the horizontal acceleration $acc^x$ and vertical velocity $v^y$ of the BV:

$$a = action = (acc^x, v^y) \tag{23}$$

Therefore, a testing scenario is modeled as:

$$\mathbb{S} = \left( s_0, (acc_0^x, v_0^y), \cdots, (acc_{k-1}^x, v_{k-1}^y) \right), \tag{24}$$

where $s_0$ is the initial state and $(acc_0^x, v_0^y), (acc_1^x, v_1^y), \cdots, (acc_{k-1}^x, v_{k-1}^y)$ is the action sequence of the BV. After analyzing HighD and NGSIM, we obtained the distribution and density of these parameters, as shown in Figure 9.Then, their value range was determined and shown in Table 4, where the discrete precision of velocity, distance, and acceleration in the x-axis was also 1 m/s, 1 m, and 0.1 m/s$^2$. In contrast, the relative distance and velocity in the y-axis was 0.1 m and 0.1 m/s.

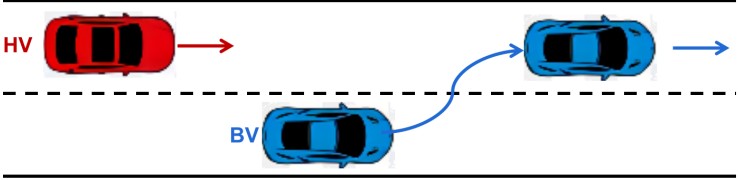

**Figure 8.** An illustration of cut-in case.

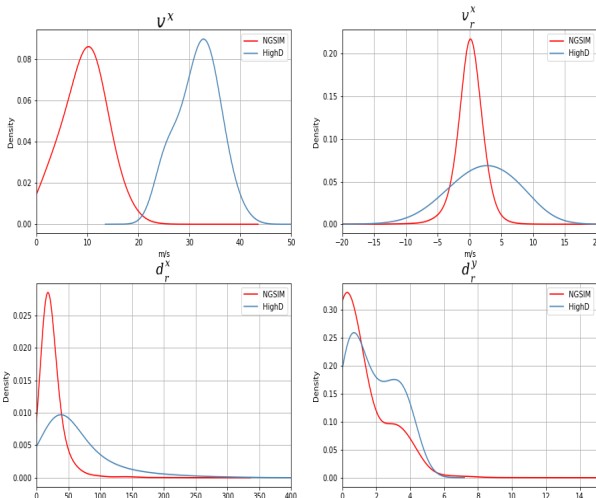

**Figure 9.** An illustration to show the distribution and density of parameters in the cut-in case based on the HighD and NGSIM.

**Table 4.** Value range of parameters in cut-in case.

| Parameter (HighD) | $v^x$ | $v_r^x$ | $d_r^x$ | $d_r^y$ | $acc^x$ | $v^y$ |
|---|---|---|---|---|---|---|
| **unit** | **(m/s)** | **(m/s)** | **(m)** | **(m)** | **(m/s²)** | **(m/s)** |
| maximum | 42 | 16 | 345 | 5 | 1.9 | 1.8 |
| minimum | 23 | −19 | 0 | 0 | −1.1 | 0.1 |
| **Parameter (NGSIM)** | $v^x$ | $v_r^x$ | $d_r^x$ | $d_r^y$ | $acc^x$ | $v^y$ |
| **unit** | **(m/s)** | **(m/s)** | **(m)** | **(m)** | **(m/s²)** | **(m/s)** |
| maximum | 29 | 17 | 224 | 13 | 3.4 | 2.0 |
| minimum | 0 | −15 | 2 | 0 | −3.4 | 0.0 |

As in the previous experiment, based on our preprocessed data and modeling method, the $\mathcal{E}(s,a)$ of cut-in case was designed as:

$$
\begin{aligned}
\mathcal{E}(s,a) &= P(a|s) \\
&= P\Big((acc^x, v^y)|(v^x, v_r^x, d_r^x, d_r^y)\Big).
\end{aligned}
\tag{25}
$$

$\mathcal{S}(s,a)$ was also measured according to the result of $\mathcal{TTC}[(s,a)]$. Then, through multiple iterations, the $\mathcal{ESC}$ matrix was obtained. According to the $\mathcal{ESC}$ matrix, we selected the initial state and action based on the threshold 0. We verified these generated scenarios with simulation experiments using the IDM model.

## 4. Analysis and Discussion

Figure 7 shows that the accident rate of the scenarios generated by the value of $\mathcal{ESC}$ was much higher than that selected by the possibility. (a) and (b) depict the accident rate of simulation experiments in the selected scenarios library obtained for the car-following case using the two datasets. The green curve shows the accident rate in the chosen scenarios using the $\mathcal{ESC}$ algorithm, and the blue curve shows the accident rate in the chosen scenarios just by the possibility of occurrence. The accident rate of (a) showed the most significant increase, with a 22.6% increase over 0.07%, which is more than 300. (b) shows that the accident rate of the test scenarios generated by this method is about 35%. (c) and (d) show the accident rate in the cut-in case. The accident rates selected by $\mathcal{ESC}$ were about 40% and 34%, much higher than those selected by the blue. Therefore, this method can generate challenging scenarios in both the car-following and cut-in cases.

When comparing these four pictures horizontally, it is clear that the larger the space, the lower the accident rate in the same case using different datasets, which aligns with cognition. However, by comparing the two pictures using the same dataset, the cut-in case with a larger modeling space is shown to have a similar or higher accuracy. This problem is related to our simulation model IDM. For example, in HighD, IDM only focuses on horizontal interactions, although the space of the cut-in case is:

$$
\begin{aligned}
size(state) &= 20 \times 36 \times 346 \times 18 \times 6 \\
&= 2.69 \times 10^7, \\
size(action) &= 31 * 23 = 713, \\
size(\mathcal{ESC}\ matirx) &= 1.92 \times 10^{10},
\end{aligned}
\tag{26}
$$

This is much larger than the space of the car-following case. The horizontal space of the cut-in case is:

$$
size(horizon) = 20 \times 36 \times 346 \times 31 = 7.72 \times 10^6,
\tag{27}
$$

This is smaller than the space of the car-following case.

In addition to accuracy, the method's coverage should be considered. Coverage, like recall in classification problems, ensures that high-risk scenarios are selected, whereas accuracy, like precision, focuses on selecting high-risk scenarios. First, we computed the *esc* of the accident scenarios chosen from the control experiments, and their *esc* values were all greater than the threshold of 0. This means that these scenarios are part of the scenario library chosen by *esc*. Second, theoretical analysis shows that each state and action was assigned the *esc* valve and occurred in a specific space. The BV decision was based on the current state. The *esc* value of the final accident state must exceed the threshold, and the influence can be passed to the previous state. As a result, this is reasonable to ensure coverage. The quantification method based on functional safety is practical and feasible in the testing scenario generation problem due to its high accuracy and coverage.

## 5. Conclusions

In this paper, we propose a practical quantitative evaluation approach to generate challenging scenarios for intelligent vehicle testing. The approach focused on existing problems, including that the existing methods to generate testing scenarios lack uniform and popular standards. They do not solve quantification problems, making it difficult to use them in industry. Our approach provides a highly operable standardized testing process.

Our method uses the three factors of risk, provided in ISO 26262—exposure, severity, and controllability–to quantify the challenge of a testing scenario. Experiments were conducted to apply this method to a car-following case and cut-in case to investigate the effectiveness of the proposed approach. The results show that it is possible to assess the risk of scenarios and select challenging scenarios to test intelligent vehicles by quantifying the risk of state and action using reinforcement learning based on functional safety. The method can ensure both accuracy and coverage. Therefore, quantitative risk evaluation enables efficient testing, reducing the intelligence's validation consumption.

Other cases, such as overtaking and crossing, can be considered in future work. The number of background vehicles can also be increased. As the number of traffic participants increases, the interaction factors will rapidly increase, and the scenario space will quickly become more extensive. Although the currently applied reinforcement learning methods are insufficient to take on this vast space, the quantification approach using ISO 26262 can be implemented by other algorithms that can support multiple parameters, such as deep reinforcement learning (DRL).

**Author Contributions:** Writing—original draft, K.M. and R.Z.; writing—review and editing, Z.L. and K.Z. All authors have read and agreed to the published version of the manuscript.

**Funding:** This research was supported by the key-Area Research and Development Program of Guangdong Province (2020B0909050003) and Science and Technology Innovation Committee of Shenzhen (CJGJZD20200617102801005).

**Institutional Review Board Statement:** Not applicable.

**Informed Consent Statement:** Not applicable.

**Data Availability Statement:** The original contributions presented in the study are included in the article, and further inquiries can be directed to the corresponding author.

**Acknowledgments:** The authors would like to thank the reviewers and editors for improving this manuscript, the key-Area Research and Development Program of Guangdong Province (2020B0909050 003), and Science and Technology Innovation Committee of Shenzhen (CJGJZD20200617102801005).

**Conflicts of Interest:** The authors declare no conflict of interest.

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
