# Peer review of "A Quantitative Approach of Generating Challenging Testing Scenarios Based on Functional Safety Standard"

_applsci, doi:10.3390/app13063494_

Round 1

Reviewer 1 Report

The authors developed an algorithm to create different and challenging driving test scenarios for intelligent and autonomous driving. The authors have discussed their subject comprehensively by getting some important things presented in some figures and their contributions to the literature are nice. The results and discussion section was written very well. So, I recommend for Publication “Acceptable with Minor Revision”.

Necessary changes suggested to be made are listed below.

1. The “Abstract” was not written well, and I cannot exactly understand what the authors intended. Please give your motivation, key contributions and key results.

2. I asked the authors to clarify in the introduction the importance of the research, mention the controversy to solve and mention the highlights.

3. In the equation, the authors should use subscripts or superscripts to define the variables. For example; In Equations 22-25 or Table 4, the variables were written as “x_v”, “x_r_v” etc. This type of variable definition makes it difficult to read and understand. Please, check the all text to rearrange this type of definition.

4. The texts in the figures are too small (For example; Figure 9 and Figure 10). It is really hard to read them. Additionally, the format of the text in the figures are different from each other. Please use only one format and text size for all graphs. Please check the all graphs and figures to increase the readability of the paper.

5. Authors should edit the English of the paper. Some spell checks required for all text. 

Author Response

Many thanks to you for encouraging our work and for giving useful comments for clarifying, improving, and correcting some materials in the paper.

Now we have carefully revised the paper according to your comments, as explained below.

All the modifications are highlighted in red in the revised manuscript.

Point 1: The “Abstract” was not written well, and I cannot exactly understand what the authors intended. Please give your motivation, key contributions, and key results.

Response 1: Thank you for your instructive suggestions. According to your helpful advice, we have added some content to the section of the abstract. The modification is between line 1 and line 13.

Point 2: I asked the authors to clarify in the introduction the importance of the research, mention the controversy to solve, and mention the highlights.

Response 2: Thank you for your careful reading of our manuscript. As you suggested, we have rewritten the relevant contents in the second and penultimate paragraphs of the introduction, including the current challenges in this field and the focus of our work. The modification is in lines 28-35 and lines 100-117.

Point 3: In the equation, the authors should use subscripts or superscripts to define the variables. For example; In Equations 22-25 or Table 4, the variables were written as “x_v”, “x_r_v” etc. This type of variable definition makes it difficult to read and understand. Please, check the all text to rearrange this type of definition.

Response 3: Thank you for your careful reading of our manuscript. We are sorry for the type of variable definition of some variables. We have carefully corrected type of definition throughout the manuscript. Thank you again.

Point 4: The texts in the figures are too small (For example; Figure 9 and Figure 10). It is really hard to read them. Additionally, the format of the text in the figures is different from each other. Please use only one format and text size for all graphs. Please check all graphs and figures to increase the readability of the paper.

Response 4: Thank you very much. We are sorry for the readability of the paper. We have carefully adjusted the format and text size in Figures 5,8,9 in the new manuscript.

Point 5:  Authors should edit the English of the paper. Some spell checks are required for all text.

Response 5: Thank you very much. We regret there were problems with the English. The paper has been carefully revised by a professional language editing service to improve grammar and readability. Thank you again.

Reviewer 2 Report

Minor Revisions

*) In the text there are some typos that should be removed.

*) Please, if possible, make all captions self-explanatory.

*) For each non-original mathematical formula at least one relevant reference should be associated.

*) Concerning Algorithm 1, what about its computational Complexity? Please, if possible, specify.

ù*) Many symbols have not been defined. Please, fill this gap.

Major Revisions

*) Its is not so clear the link between (22) and (23). Please specify in more detail.

*) What are the basic assuption of (13)? If so, have they been verified?

*) Problablym a flowchart of the proposed approach shoud be added in the paper. This would hep the read to better understand the contribution of the paper.

*) Often, data can be affected by uncertanties and/or imprecisions. In such cases, scientific literature suggests to implement tools based on fuzzy logic as data preprocessor. Obfiously, I do not ask to the Authors to implement such tools. I just ask to highlight this aspect in the introduction inserting one/two sentences listing in the bibliography the following relevant references:

doi: 10.3390/s22114232

doi: 10.1109/TIFS.2017.2707332

doi: 10.1109/JSTARS.2017.2664119

Author Response

Many thanks to you for encouraging our work and for giving useful comments for clarifying, improving, and correcting some materials in the paper.

Now we have carefully revised the paper according to your comments, as explained below.

All the modifications are highlighted in red in revised manuscript.

Minor Revisions

Point 1: In the text there are some typos that should be removed.

Response 1: Thank you very much. We regret there were problems with the typos. The paper has been carefully revised by a professional language editing service to improve grammar and readability. Thank you again.

Point 2: Please, if possible, make all captions self-explanatory.

Response 2: Thank you for your careful reading of our manuscript. As you suggested, we have rewritten some captions to describe the figures clearly. The modification is in Figures 1, 2, 4, and 5 in the new manuscript.

Point 3: For each non-original mathematical formula at least one relevant reference should be associated.

Response 3: Thank you for your instructive suggestions. As you suggested, we have added the reference to every non-original mathematical formula. 

Point 4: Concerning Algorithm 1, what about its computational Complexity? Please, if possible, specify.

Response 4: Thank you very much. We have added reference 54 as the complexity study basis of Algorithm 1. And We have provided the memory complexity and computation complexity of each iteration in lines 236-239.

Point 5: Many symbols have not been defined. Please, fill this gap.

Response 5: Thank you very much. We have carefully rechecked symbols and added the definition.

Major Revisions

Point 6: It is not so clear the link between (22) and (23). Please specify in more detail.

Response 6: Thank you very much. We are sorry for not describing the link between (22) and (23) clearly. According to your helpful advice, we have rewritten the two relevant paragraphs in lines 307-327 to describe (22), and (23) and the link between them.

Point 7: What are the basic assuption of (13)? If so, have they been verified?

Response 7: Thank you very much. (13) is the equation of the intelligent driver model (IDM) to model the traffic flow behavior over varied traffic conditions, proposed and verified in reference 53. As shown in reference 49, IDM is deemed as a potential car-following model in coding following behavior for autonomous vehicles in many works and is widely used in relevant fields. Thank you again.

Point 8: Problablym a flowchart of the proposed approach should be added to the paper. This would help the readers to better understand the contribution of the paper.

Response 8: Thank you for your instructive suggestions. We have added Figure 3 to show the flowchart of the proposed method.

Point 9: Often, data can be affected by uncertainties and/or imprecision. In such cases, scientific literature suggests to implement tools based on fuzzy logic as data preprocessors. Obviously, I do not ask the Authors to implement such tools. I just ask you to highlight this aspect in the introduction inserting one/two sentences listing in the bibliography the following relevant references:

doi: 10.3390/s22114232

doi: 10.1109/TIFS.2017.2707332

doi: 10.1109/JSTARS.2017.2664119

Response 9: Thank you for your instructive suggestions. As an important method for data pre-processing, fuzzy logic is often used in the process of test scenario generation and selection methods. We have already mentioned this and cited references 55, 56, and 57 in line 102. Thank you again.

Round 2

Reviewer 2 Report

All suggestions have been implemented. Therefore, in my opinion, the paper deserves publications.